# *Escherichia coli* Nissle 1917 Enhances Efficacy of Oral Attenuated Human Rotavirus Vaccine in a Gnotobiotic Piglet Model

**DOI:** 10.3390/vaccines10010083

**Published:** 2022-01-06

**Authors:** Husheem Michael, Ayako Miyazaki, Stephanie N. Langel, Joshua O. Amimo, Maryssa K. Kick, Juliet Chepngeno, Francine C. Paim, David D. Fischer, Gireesh Rajashekara, Linda J. Saif, Anastasia N. Vlasova

**Affiliations:** 1Center for Food Animal Health, Department of Animal Sciences, Ohio Agricultural Research and Development Center, The Ohio State University, Wooster, OH 44691, USA; michael.332@osu.edu (H.M.); miyaan@affrc.go.jp (A.M.); stephanie.langel@duke.edu (S.N.L.); amimo.3@osu.edu (J.O.A.); kick.28@osu.edu (M.K.K.); chepngeno.1@osu.edu (J.C.); franchimelo@gmail.com (F.C.P.); fischedd@udmercy.edu (D.D.F.); rajashekara.2@osu.edu (G.R.); saif.2@osu.edu (L.J.S.); 2Division of Viral Disease and Epidemiology, National Institute of Animal Health, National Agriculture and Food Research Organization, Tsukuba 305-0856, Ibaraki, Japan; 3Duke Human Vaccine Institute, Duke University School of Medicine, Durham, NC 27707, USA; 4Division of Integrated Biomedical Sciences, University of Detroit Mercy School of Dentistry, Detroit, MI 48208, USA

**Keywords:** probiotics, live attenuated rotavirus vaccine, human rotavirus infection, innate immunity, adaptive immunity, gnotobiotic pigs

## Abstract

Human rotavirus (HRV) infection is a major cause of viral gastroenteritis in young children worldwide. Current oral vaccines perform poorly in developing countries where efficacious vaccines are needed the most. Therefore, an alternative affordable strategy to enhance efficacy of the current RV vaccines is necessary. This study evaluated the effects of colonization of neonatal gnotobiotic (Gn) pigs with *Escherichia coli* Nissle (EcN) 1917 and *Lacticaseibacillus rhamnosus* GG (LGG) probiotics on immunogenicity and protective efficacy of oral attenuated (Att) HRV vaccine. EcN-colonized pigs had reduced virulent HRV (VirHRV) shedding and decreased diarrhea severity compared with the LGG-colonized group. They also had enhanced HRV-specific IgA antibody titers in serum and antibody secreting cell numbers in tissues pre/post VirHRV challenge, HRV-specific IgA antibody titers in intestinal contents, and B-cell subpopulations in tissues post VirHRV challenge. EcN colonization also enhanced T-cell immune response, promoted dendritic cells and NK cell function, reduced production of proinflammatory cytokines/Toll like receptor (TLR), and increased production of immunoregulatory cytokines/TLR expression in various tissues pre/post VirHRV challenge. Thus, EcN probiotic adjuvant with AttHRV vaccine enhances the immunogenicity and protective efficacy of AttHRV to a greater extent than LGG and it can be used as a safe and economical oral vaccine adjuvant.

## 1. Introduction

Human rotavirus (HRV) is the leading cause of viral gastroenteritis in infants and children that results in significant morbidity and mortality, especially in developing countries [1,2,3]. There are two types of oral HRV vaccines available: RotaTeq^®^ (Kenilworth, NJ, USA) (pentavalent) and Rotarix^®^ (Brentford, UK) (monovalent). Studies have shown that there is reduced efficacy of HRV and other vaccines in developing countries, which presents a major concern and a global health challenge [4,5,6,7]. Many factors, such as malnutrition, micronutrient deficiencies, and breastfeeding have been suggested to be involved in the reduced efficacy of enteric vaccines in developing countries [8,9,10]. Therefore, alternative strategies are needed to improve HRV and efficacy of other enteric vaccines.

Commensal/probiotic colonization and their beneficial effects play a vital role early in life that may result in lifelong benefits [11]. Thus, early colonization by probiotic species may promote gut immunity maturation to enhance oral vaccine efficacy and moderate the severity of enteric infections. Probiotics have also been used to improve immune responses to oral vaccines, ameliorate gastrointestinal infections/inflammation and inflammatory bowel disease in children, and impede *Helicobacter pylori* replication, allergies, and cancer [12,13,14,15,16,17,18,19,20,21,22,23]. *Escherichia coli* Nissle (EcN) 1917 is a Gram-negative probiotic that was shown to be highly efficient in protecting gnotobiotic (Gn) piglets against HRV infection and disease by modulating innate and adaptive immunity, and directly protecting the intestinal epithelium by binding to HRV particles [24,25]. *Lacticaseibacillus rhamnosus* GG (LGG) is a Gram-positive probiotic commonly used to treat or prevent RV diarrhea in children. Prophylactic supplementation of LGG to children significantly reduced the incidence of HRV disease [26]. Furthermore, *Lacticaseibacillus* strains were demonstrated to significantly reduce diarrhea severity in hospitalized children [27]. Previous studies also demonstrated that colonization of Gn piglets with  LGG and *Bifidobacterium lactis* resulted in a significant reduction in both fecal HRV shedding titers and diarrhea severity in the Gn piglet model of HRV infection [12].

Gn pigs are immunocompetent at birth but immunologically immature [28]. They are ideal to study HRV pathogenesis due to their susceptibility to HRV infection, as well as the greater anatomic, physiological, and immunological similarities between pigs and children [29]. Gn piglets are caesarian-obtained and retained in antiseptic isolators to ensure their sterilized status, allowing analyses of gut colonization with specific bacteria, and a defined or fecal microbiota. Hence, Gn piglets are a distinctive animal model to examine host metabolism, neonatal immune responses, gastrointestinal viral infections, and oral vaccines [30,31].

In this study, we investigated the role of selected probiotics (EcN and LGG), on adaptive and innate immune responses to live oral attenuated (Att) HRV Wa (G1P [8]) vaccine (genotypically identical to Rotarix vaccine strain, G1P [8]) and on protection against virulent (Vir) HRV Wa infection. We hypothesized that initial colonization by the probiotics would enhance immune maturation and responses (adjuvant properties) to an oral (AttHRV) vaccine, thereby reducing VirHRV diarrhea severity (as prophylactic properties). Improving our understanding of the adjuvant properties of probiotics will provide a novel means to enhance efficacy of other vaccines, as well as ameliorate viral gastroenteritis not only in infants but also in piglets and calves. Such innovation is needed especially in developing countries where HRV vaccine use is inaccessible, expensive, impeded, or ineffective. Our results demonstrated that the probiotic EcN increased efficacy of oral AttHRV vaccine by increasing both innate and adaptive immune responses to protect against VirHRV challenge.

## 2. Materials and Methods

### 2.1. Virus

The cell culture adapted attenuated HRV (AttHRV) Wa G1P [8] strain passaged in African green monkey kidney cells (MA-104) was used as a vaccine at a dose of 1 × 10^7^ fluorescent foci-forming units (FFU) [32]. Virulent HRV (VirHRV) Wa strain passaged 25–26 times in Gn piglets was used to orally challenge piglets at a dose of 1 × 10^6^ FFU as described previously [33,34].

### 2.2. Animal Experiments

This study was approved by the Ohio State University Institutional Animal Care and Use Committee. Piglets were derived from near-term sows (Landrace × Yorkshire × Duroc cross-bred) by hysterectomy and maintained in sterile isolators as described previously [35]. All the piglets were fed with ultrahigh-temperature processed commercial bovine milk diet throughout the experiment. *Lacticaseibacillus rhamnosus* GG (LGG) a Gram-positive probiotic, was used in these experiments for comparison with the Gram-negative probiotic *Escherichia coli* Nissle (EcN) 1917 [12,13,36]. Neonatal piglets were randomly assigned to two groups (Figure 1): EcN/AttHRV/VirHRV (*n* = 7) and LGG/AttHRV/VirHRV (*n* = 6). EcN and LGG inoculum was prepared as described previously [24] and sequentially colonized orally at day 3 of age at a dosage of 10^5^ CFU/pig/timepoint in 0.1% peptone water. All pigs were orally vaccinated twice at 7 day intervals with AttHRV on post bacterial colonization day (PBCD) 3/post first vaccination day (PVD) 0, and PBCD 10/PVD 7(0) (post second vaccination day 0, hereafter referred to as PVD7 (0)). At PBCD 17/PVD 14 (7)/post challenge day (PCD) 0, a subset of pigs from each of the group was euthanized to assess immune vaccine responses pre challenge. The remaining pigs were challenged with homologous VirHRV Wa and euthanized at PBCD 24/PVD 21 (14)/PCD 7. All piglets were challenged with VirHRV at a dose of 1 × 10^6^ FFU per piglet at PBCD 17 and were euthanized at PBCD 24/post VirHRV challenge day (PCD) 7. Systemic and intestinal tissues were collected to isolate mononuclear cells (MNCs) for subsequent immunological assays. Serum and intestinal contents were collected to determine the HRV specific and total antibody responses.

### 2.3. Assessment of Clinical Signs

Rectal swabs were collected daily after VirHRV challenge. Fecal consistency was scored as follows: 0, normal; 1, pasty/semi-liquid; 2, liquid; pigs with fecal scores more than 1 were considered diarrheic. Rectal swabs were suspended in 2 mL of MEM (Life Technologies, Waltham, MA, USA) supplemented with antibiotics/antimycotic, clarified by centrifugation at 800× *g* for 10 min at 4 °C, and stored at −20 °C until quantification of infectious HRV by a cell culture immunofluorescence (CCIF) assay.

### 2.4. Isolation of Mononuclear Cells (MNCs) and Immunological Assays

Systemic (blood, spleen) and intestinal (ileum, duodenum) tissues were collected to isolate MNCs as described previously [29,34,37,38,39]. The purified MNCs were suspended in E-RPMI 1640. The viability of each MNC preparation was determined by trypan blue exclusion (≥95%).

The HRV antibody and total immunoglobulin (Ig) isotype titers in serum, SIC, and LIC were detected by ELISA. To determine the intestinal antibody responses, SIC and LIC were collected with protease inhibitors in the medium. MNCs isolated from blood, spleen, duodenum, and ileum were analyzed by ELISPOT assay to quantitate numbers of HRV-specific antibody secreting as described previously [34,39,40].

### 2.5. Serum Cytokines

Serum samples were collected and analyzed for proinflammatory (IL-6, IL-17, TNF-α, IFN-γ) and T regulatory (IL-10, TGF-β) cytokine levels as described previously with modifications [32,38].

### 2.6. Flow Cytometry Analysis

Freshly isolated MNCs were stained to determine frequencies of T-cell subsets: T helper cells (CD3^+^CD4^+^), cytotoxic T cells (CD3^+^CD8^+^), natural T regulatory cells (CD4^+^/CD8^+^CD25^+^FOXP3^+^), activated T regulatory cells (CD4^+^/CD8^+^CD25^+^FOXP3^−^), and inducible T regulatory cells (CD4^+^/CD8^+^CD25^−^FOXP3^+^) [38,40]. MNCs were stained to assess frequencies of conventional dendritic cells (DCs) (cDCs, SWC3a^+^CD4^−^CD11R1^+^) and plasmacytoid DCs (pDCs, SWC3a^+^CD4^+^CD11R1^−^), activated cDC (SWC3a^+^CD4^−^CD11R1^+^MHC II^+^), activated pDC (SWC3a^+^CD4^+^CD11R1^−^MHC II^+^), and CD103^+^ cDC (SWC3a^+^CD4^−^) and CD103^+^ pDC (SWC3a^+^CD4^+^) marker expression on DCs and Toll-like receptor (TLR) expression on MNCs as reported previously [13,24,41,42]. TLR3 (ligand of double-stranded RNAs), TLR4 (ligand of bacterial lipopolysaccharide), and TLR9 (ligand of bacterial CpGs) were examined in our experiments. Similarly, frequencies of resting/memory antibody forming B cells (CD79β^+^CD2^−^CD21^−^), Ig-secreting B cells (CD79β^+^CD2^+^CD21^−^), naïve antibody-forming B cells (CD79β^+^CD2^+^CD21^+^), and activated antibody-forming B cells (CD79β^+^CD2^−^CD21^+^) among systemic and intestinal CD79β^+^ B cells were analyzed using gating strategies as determined previously [13,24,43]. Frequencies of NK cells (SWC3a^+^CD16^+^) were assessed among systemic and intestinal MNCs. The frequencies and tissue distribution of apoptotic MNCs were assessed with an annexin V apoptosis detection kit. APC and propidium iodide staining solution (eBioscience) were used to discriminate apoptotic and necrotic MNCs. Analysis and gating strategies were performed as described previously [24]. Appropriate isotype-matched control antibodies were included in all analyses. Subsequently, 50,000 events were acquired per sample using BD Accuri C6 flow cytometer (BD Biosciences, San Jose, CA, USA). Data were analyzed using C6 flow sampler software. Lastly, for NK cell cytotoxicity assessment, blood MNCs and K562 cells were used as effector and target cells, respectively. Effector–target cell ratios of 10:1, 5:1, 1:1, and 0.5:1 were used, and the assay was done as described previously [38,44].

### 2.7. Isolation of Intestinal Epithelial Cells (IECs), Extraction of RNA, and Analysis of CgA, MUC2, PCNA, SOX9, and Villin Gene Expression

The IECs were isolated from jejunum (mid gut) using a modified protocol adapted from Paim et al. [25,34,45]. The viability and numbers of IECs were determined by the trypan blue exclusion method (70–80%). qRT-PCR was performed using equal amounts of total RNA (75 ng) with Power SYBR Green RNA-to-CT 1 step RT-PCR kit (Applied Biosystems, Foster, CA, USA). The gene-specific primers for enteroendocrine cells (chromogramin A (CgA)), goblet cells (mucin 2 (MUC2)), transient amplifying progenitor cells (proliferating cell nuclear antigen (PCNA)), intestinal epithelial stem cells (transcription factor SRY-box9 (SOX9)), enterocytes (villin), and β-actin were based on previously published data [25,46,47,48]. Relative gene expressions of CgA, MUC2, PCNA, SOX9, and villin were normalized to β-actin and expressed as fold change using the 2^−ΔΔCt^ method [49].

### 2.8. Statistical Analysis

All statistical analyses were performed using GraphPad Prism version 6 (GraphPad Software, Inc., La Jolla, CA, USA). Log_10_ transformed isotype ELISA antibody titers were analyzed using one-way ANOVA followed by Duncan’s multiple range test. Correlation analysis was performed using nonparametric correlation Spearman’s method (Appendix A). Data represent the mean numbers of HRV specific antibody secreting cells per 5 × 10^5^ MNCs (* *p* < 0.05, ** *p* < 0.01, and *** *p* < 0.001). Values or error bars indicate the standard error of the mean.

## 3. Results

### 3.1. EcN Colonization Reduced the Fecal Scores and HRV Shedding Post HRV Challenge

Diarrheal scores and HRV fecal shedding were recorded daily post VirHRV challenge. Analysis revealed that EcN colonization resulted in greater protection from diarrhea (Table 1). Moreover, EcN-colonized pigs had reduced mean cumulative fecal score (6.1), delayed onset (5.3), and shortened duration of diarrhea (0.4) compared with LGG colonized pigs (7.0, 4.0, and 1, respectively).

HRV shedding analysis revealed that 42.9% of EcN pigs shed compared with 66.7% LGG-colonized pigs, and the reduced shedding coincided with reduced diarrheal scores (Table 1). Furthermore, a reduction in HRV viral shedding titers was observed in EcN (GMT = 100 FFU/mL) compared with LGG-colonized pigs (GMT = 171 FFU/mL). Although EcN-colonized pigs had greater protection against diarrhea and lower HRV shedding titers, the mean day of onset of HRV shedding was earlier than the LGG group; however, there was no significant difference in duration of HRV shedding between the EcN- and LGG-colonized groups (1.1 vs. 0.8 days, respectively).

### 3.2. HRV-Specific Iga Antibody Titers and Antibody Secreting Cells (ASCs)

At PVD14(7) and pre challenge day 0, EcN colonization enhanced the effects of AttHRV vaccination, i.e., increasing the HRV-specific IgA and IgG antibody titers in serum: IgA and IgG ASCs in systemic and intestinal MNCs, and IgM ASCs in intestinal MNCs compared with LGG colonization (Table 2, Appendix A). However, comparable titers were observed in SIC while titers were not available in LIC in both groups. Moreover, EcN colonization increased total immunoglobulin concentrations in serum: total IgA IgSCs in ileum and spleen, and total IgG/IgM IgSCs in spleen and duodenum (Appendix A).

Post VirHRV challenge (PCD7), coincident with reduced fecal scores and HRV shedding titers, EcN colonization increased the HRV specific IgA antibody titers in serum, SIC, and LIC, as well as HRV-specific IgA ASCs in systemic, and intestinal MNCs compared with LGG colonized pigs (Table 2, Appendix A). Similar trends were observed for HRV-specific IgG and IgM antibody titers in serum: IgG ASCs in splenic and ileal MNCs, IgM ASCs in intestinal MNCs, and total immunoglobulin concentrations in serum (Appendix A, Appendix A). Moreover, EcN colonization increased the total IgA IgSCs in ileum and spleen: IgG IgSCs in ileum, spleen, and duodenum, and IgM IgSCs in spleen (Appendix A). These results indicate that EcN colonization contributed to the increased AttHRV vaccine-induced antibody titers and ASCs pre challenge and further potentiated secondary immune responses following VirHRV challenge.

### 3.3. Effects of the Probiotic Colonization on Frequencies of B Lymphocyte Phenotypes

At PVD14(7) and pre challenge day 0, EcN compared with LGG colonization decreased the frequencies of B-cell subpopulations (activated, naïve (except duodenum), resting/memory, Ig-secreting, IgA^+^) in tissues (Table 2). Interestingly and contrary to pre challenge results, coinciding with increased HRV-specific ASCs, antibody titers, reduced fecal scores, and HRV shedding titers om EcN colonized pigs, post VirHRV challenge (PCD7) had increased frequencies of all the above B-cell subpopulations in tissues (Table 2, Appendix A). These findings indicate that the AttHRV vaccine primed the B-cell immune responses that resulted in reduced clinical parameters following VirHRV challenge.

### 3.4. EcN Colonization Altered the Frequencies of T Helper and Cytotoxic T Cells in Systemic and/or Intestinal MNCs Pre/Post VirHRV Challenge

LGG colonization resulted in slightly elevated frequencies (percent) of T helper cells (CD3^+^CD4^+^) in systemic MNCs, while EcN colonization increased them in intestinal MNCs pre challenge (PCD0). However, EcN colonization increased the frequencies of cytotoxic T cells (CD3^+^CD8^+^) in systemic and ileal MNCs pre challenge compared with LGG colonization (Table 3). Post VirHRV challenge (PCD7), EcN colonization decreased the frequencies of T helper cells in the systemic and ileal MNCs, while they were increased in the duodenum compared with LGG colonization. On the other hand, frequencies of cytotoxic T cells were decreased in blood and ileum but increased in the spleen and duodenum (Table 3, Appendix A). This indicates that AttHRV vaccine efficiently primed the T-cell immune responses, reducing the severity of clinical disease following HRV challenge.

### 3.5. EcN Colonization Increased HRV Specific CD4^+^/CD8^+^ IFN-γ-Producing T Cells in Ileum Post VirHRV Challenge

At PVD14(7) and pre challenge day 0, LGG colonization increased the frequencies of HRV-specific CD3^+^CD4^+^ IFN-γ-producing T cells in the spleen and ileum. EcN colonization resulted in elevated frequencies of CD3^+^CD8^+^ IFN-γ-producing T cells in spleen but lower frequencies in ileum, suggesting a systemic Th1 associated enhancing effect of EcN on AttHRV vaccine (Table 3). Post VirHRV challenge, like pre challenge, comparable effects were observed in spleen. However, EcN colonization increased the frequencies of CD3^+^CD4^+^ IFN-γ-producing T cells in ileum. Contrary to pre challenge, EcN colonization decreased the frequencies of CD8^+^ IFN-γ-producing T cells in spleen, while it increased the frequencies in ileum post VirHRV challenge (Table 3). Thus, our results suggest that EcN enhanced the effects of AttHRV vaccine, in some cases biasing toward Th1 responses, to alleviate HRV infection.

### 3.6. EcN Colonization Altered the Frequencies of CD4^+^/CD8^+^ T Regulatory Cell (T Reg) Subpopulations in Systemic and Intestinal Mncs Pre/Post VirHRV Challenge

At PVD14(7) and pre challenge day 0, EcN colonization resulted in slightly higher frequencies of activated CD4^+^ T regs in splenic and intestinal MNCs, activated CD8^+^ T regs in systemic and ileal MNCs, inducible CD4^+^/CD8^+^ T regs in systemic and intestinal MNCs, and natural CD4^+^/CD8^+^ T regs in spleen and ileum (Table 3) consistent with promotion of immune homeostasis by EcN after AttHRV vaccine.

Post VirHRV challenge, EcN colonization increased the frequencies of activated CD4^+^ T regs in splenic and intestinal MNCs, activated CD8^+^ T regs in spleen and duodenum, inducible CD4^+^ T regs in systemic and duodenal MNCs, inducible CD8^+^ T regs in spleen, natural CD4^+^ T regs in splenic and intestinal MNCs, and natural CD8^+^ T regs in systemic and intestinal MNCs (Table 3). These results indicate that EcN enhanced the ability of the AttHRV vaccine to prime T reg immune responses pre challenge that resulted in reduced diarrheal severity and HRV shedding following HRV challenge.

### 3.7. EcN Colonization Reduced Proinflammatory and Increased Immunoregulatory Cytokine Level in Serum Pre/Post VirHRV Challenge

Coinciding with decreased fecal scores and HRV shedding titers, EcN colonization pre challenge and post VirHRV challenge resulted in significantly reduced proinflammatory (IL-6, IL-17, TNF-α, IFN-γ) cytokine levels in serum in most cases compared with LGG colonization (Figure 2a,b). In contrast, EcN colonization significantly increased the levels of immunoregulatory (IL-10, TGF-β) cytokines compared with LGG colonization. These data suggest that EcN reduced local (gut) inflammation caused by HRV infection.

### 3.8. EcN Colonization Altered the Frequencies of cDCs, pDCs, CD103^+^ cDCs/pDCs, and Activated cDCs/pDCs in Systemic and Intestinal Tissues Pre/Post VirHRV Challenge

At PVD14(7) and pre challenge day 0, EcN colonization increased the frequencies of cDCs in splenic and intestinal MNCs, pDCs in systemic and ileal MNCs, CD103^+^ cDCs/pDCs in intestinal MNCs, activated cDCs in intestinal MNCs, and activated pDCs in duodenum (Table 4).

Post VirHRV challenge, EcN colonization increased the frequencies of cDCs in both systemic and intestinal MNCs, pDCs in systemic and ileal MNCs, CD103^+^ cDCs in spleen and ileum, CD103^+^ pDCs in spleen and duodenum, activated cDCs in systemic MNCs, and activated pDCs in blood (Table 4, Appendix A). The data suggest that EcN colonization could enhance innate immune signaling against VirHRV that led to enhanced HRV-specific IgA ASCs and antibody titers, improved epithelial barrier [25], and reduced diarrhea severity and HRV shedding titers.

### 3.9. EcN Colonization Reduced the Frequencies of Proinflammatory and Increased Immunoregulatory Toll-Like Receptor (TLR)-Expressing MNCs

At PVD14(7) and pre challenge day 0, EcN colonization decreased the frequencies of TLR4 (associated with proinflammatory signaling)-expressing MNCs in systemic and systemic/intestinal tissues, respectively. On the other hand, EcN colonization increased the frequencies of TLR3/TLR9 (associated with RV protection and anti-inflammatory signaling)-expressing MNCs in systemic and intestinal tissues (Table 4).

Post VirHRV challenge, EcN colonization decreased the frequencies of TLR4-expressing MNCs in intestinal and systemic/intestinal tissues, coinciding with decreased diarrheal severity and HRV shedding titers. TLR3-expressing MNCs were increased in systemic and intestinal tissues, while TLR9-expressing MNCs were increased in the ileum of EcN-colonized pigs (Table 4, Appendix A). These data indicate that EcN colonization upregulated TLR3/9 expression pre challenge that enhanced HRV-specific antibody production and decreased proinflammatory responses against HRV infection.

### 3.10. EcN Colonization Altered the Frequency of NK Cells, Apoptotic MNCs, and NK Function Pre/Post VirHRV Challenge

At PVD14(7) and pre challenge day 0, EcN colonization increased the frequencies of NK cells in blood and ileum MNCs (Table 4). However, post VirHRV challenge, EcN colonization increased the frequencies of NK cells in systemic and duodenal MNCs.

Frequencies of apoptotic MNCs were decreased in intestinal MNCs pre challenge (Table 4). On the other hand, frequencies of apoptotic MNCs decreased in EcN colonized pigs in systemic and ileal MNCs post VirHRV challenge.

NK cell function in blood MNCs was enhanced in EcN-colonized pigs compared with LGG-colonized pigs both pre/post VirHRV challenge (Figure 2c,d). This suggests that EcN colonization promoted innate immune responses pre challenge, thereby improving protection against VirHRV challenge.

### 3.11. EcN Colonization Upregulated the Expressions of Villin, SOX9, PCNA, MUC2, and CgA mRNA Levels in Jejunal Epithelial Cells Post VirHRV Challenge

Gene expression levels of villin, SOX9, PCNA, MUC2, and CgA mRNA levels were assessed in jejunal epithelial cells. The relative mRNA levels of all the genes were increased in jejunal epithelial cells of EcN compared with LGG colonized pigs (Appendix A). This coincided with the decreased severity of HRV shedding and diarrhea. This suggests that EcN increased numbers of enterocytes and other epithelial cells and may have improved probiotic adherence post VirHRV challenge.

## 4. Discussion

Probiotic intervention is a potential strategy to immunomodulate adaptive and innate immunity to enteric infections and/or enhance efficacy of vaccines. Probiotic investigations in infants and children using vaccines have focused mainly on antibody responses, and a small number of studies have addressed T-cell responses [14]. Using the relevant neonatal piglet model of HRV pathogenesis, we previously demonstrated the impact of two probiotics (EcN (Gram-negative probiotic) and LGG (Gram-positive probiotic)) on immune responses following VirHRV challenge; however, in this study we determined the impacts of these probiotics on innate and adaptive immune responses following oral 2× AttHRV vaccination and pre/post VirHRV challenge.

Our study showed that EcN colonization resulted in significant reduction in HRV shedding, diarrheal scores, delayed onset, and duration of diarrhea compared with previously published observations [24] suggesting that EcN can act as an adjuvant for live oral AttHRV vaccines.

Our results suggest that the EcN probiotic, which effectively colonized the Gn piglets, beneficially immunomodulated B cell responses to oral AttHRV vaccine pre challenge. Interestingly in this study, HRV-specific IgA/IgG ASCs were elevated in both ileal and duodenal MNCs compared to our previous study where the effect was observed only in duodenum post VirHRV challenge [24] owing to the enhanced efficacy of EcN on AttHRV vaccine, which resulted in substantial reduction of fecal scores and HRV shedding titers. Similarly, greatly increased total IgA IgSCs were observed in duodenum compared with our previous observation [24]. Moreover, total IgG/IgM IgSCs were undetectable in the previous study. These results indicate that EcN probiotic improved the efficacy of the live AttHRV vaccine that further enhanced long-lasting HRV antibody producing cell frequencies in EcN-colonized and VirHRV-challenged Gn pigs [29,50].

EcN colonization enhanced total and HRV-specific IgA, IgG, and IgM antibody titers in serum and HRV specific IgA antibody titers in intestinal contents which coincided with reduced diarrheal scores and HRV shedding titers. Moreover, HRV-specific/total IgA antibody responses were correlated with HRV-specific/total IgG and IgM antibody responses, further confirming the enhancement of immune responses pre-challenge against HRV infection [29,51,52]. In addition, EcN increased HRV-specific IgA Ab titers in LIC and total IgM antibody responses post VirHRV challenge, which was not observed in our previous study [24], indicative of adjuvant properties of EcN on AttHRV vaccine.

In this study, resting/memory antibody-forming B cells were enhanced in systemic and intestinal MNCs post VirHRV challenge, contrary to a previous study where these cells were enhanced only in spleen post VirHRV challenge [24], suggesting that EcN enhanced effects of AttHRV vaccine. Moreover, EcN colonization increased the frequencies of other subpopulations of B cells in tissues post VirHRV challenge, suggesting that adjuvant properties of EcN with AttHRV vaccine may have potentiated intestinal B-cell development and, thus, also increased systemic responses. Moreover, EcN colonization enhanced AttHRV vaccine efficacy by increasing activated CD4^+^ Tregs in duodenum post VirHRV and natural CD8^+^ Tregs in ileum pre challenge. This suggests that EcN colonization pre challenge enhanced T-cell immunity and induced an anti-inflammatory environment, thereby inhibiting proinflammatory immune responses post VirHRV challenge. This also implies that blockade of proinflammatory signaling by EcN adjuvant with AttHRV can represent a therapeutic approach.

Interestingly, cDCs, activated cDCs, and CD103^+^ cells were not observed in our previous studies [24] indicative of the EcN adjuvant effect on AttHRV vaccine. Moreover, pDCs were increased in duodenum post VirHRV challenge in our previous studies [24]; however, in this study, pDCs were increased in systemic and ileal MNCs pre/postVirHRV challenge, thus indicating a role of EcN in AttHRV vaccine efficacy. These results indicate that the EcN was established in the gut pre challenge and, thus, enhanced the effectiveness of AttHRV vaccine that led to the maturation of systemic and intestinal activated cDCs, promoted pDC and cDC development, and increased IgA antibody responses post VirHRV challenge in pigs [53,54,55,56].

Compared with previous observations [24] where TLR3-expressing MNC frequencies were increased in duodenum, our findings indicated that EcN colonization increased the TLR3-expressing MNCs frequencies in both systemic and intestinal MNCs pre challenge but in blood and intestinal MNCs post VirHRV challenge, suggesting that colonization may have supported immune activation of virus-induced TLR3-expressing MNCs or enhanced their persistence pre challenge, indicating a role of EcN on AttHRV effectiveness.

TLR9-expressing MNCs frequencies were increased in systemic and intestinal MNCs pre challenge and ileal MNCs post VirHRV challenge of EcN-colonized pigs; however, previously [24], TLR9-expressing MNC frequencies were increased in duodenal MNCs only, suggesting a potent beneficial effect of EcN on AttHRV vaccine. Interestingly, TLR3/9^+^ MNCs frequencies were increased in splenic (significantly) and other tissue MNCs of EcN-colonized pigs pre challenge, suggesting a possible synergistic effect of EcN colonization on AttHRV vaccine leading to upregulation of TLR3/9 expression, which contributed to the enhanced immunoglobulin responses [13,57,58]. These results indicate that EcN effects on AttHRV vaccine enhanced TLR3/9 expression and facilitated more efficient recognition of RV and dsRNA, enhanced antibodies against HRV, decreased proinflammatory cytokines/TLR4-expressing MNCs, and increased immunoregulatory cytokines pre challenge, thus improving protection against HRV diarrhea and shedding [12,32,36,59,60,61,62]. In addition, EcN acted as an immunostimulant for AttHRV vaccine that may have exerted antiviral actions via TLR signaling by modulating DC activation pre challenge.

Furthermore, NK cell function was enhanced in EcN colonized pigs pre/post VirHRV challenge, suggesting that EcN colonization produced adjuvant effects with live oral AttHRV to promote innate immune responses pre challenge that alleviated HRV infection post VirHRV challenge.

EcN mechanisms of action include direct antimicrobial impact, involvement in the bacterial–epithelial crosstalk via biofilm formation, strengthening of tight junctions of intestinal epithelial cells, interactions with immune system by causing a decrease in proinflammatory cytokines and an increase in immunoregulatory cytokines, and binding to HRV particles via histo-blood group antigen-like bacterial glycans [63]. On the other hand, LGG mechanisms of action include enhancement of barrier function by decreasing apoptosis of epithelial cells via decreased production of TNF-α, increase in production of mucin (MUC2 expression), antimicrobial effect via bacteriocin production, and enhanced mucosal immunity by increasing IgA production [64].

There is a limitation of this study in that Gn piglets without other confounding microbiota were used to determine the specific beneficial effects of EcN on AttHRV against HRV-induced immunity. Whether the findings from this study using Gn pigs can be comparable to conventional pigs or other species with a normal gut microbiota is uncertain because of the reduced immune responses in germ-free, mono-colonized or dual-colonized versus conventional animals [65,66]. However, our methodology is advantageous in terms of eliminating the confounding effects of the intestinal microbiota and defining the effects of individual or combined probiotic with AttHRV vaccine on HRV infections. Even if the magnitude of the immune responses is lower in Gn compared with conventional pigs, the overall mucosal immune responses and their kinetics are similar [12]. Our results indicate the possibility that low-cost dietary supplementation of EcN given prior to or even together with RV vaccine can protect against HRV-associated diarrhea, HRV outbreaks, and other potentially enteric infections.

In future studies, we plan to delineate the impact of EcN probiotic in pigs transplanted with a human infant fecal microbiota and vaccinated with live oral AttHRV to determine its effect pre/post VirHRV challenge. Furthermore, additional studies are needed to evaluate the efficacy of EcN efficacy on AttHRV vaccine in malnourished Gn piglets transplanted with human infant fecal microbiota, as well as in a conventional pig model.

## 5. Conclusions

Our results suggest that, compared with the Gram-positive LGG probiotic, the Gram-negative EcN probiotic had greater beneficial effects in promoting strong but balanced immunoregulatory/immunostimulatory efficacy of oral AttHRV vaccine in Gn piglets. EcN probiotic increased the effectiveness of live oral AttHRV vaccine by increasing both adaptive and innate immune responses. Our results suggest the possibility that low-cost dietary supplementation of EcN given together with RV vaccine can protect against HRV-associated diarrhea, HRV outbreaks, and other potentially enteric infections.

## Figures and Tables

**Figure 1 vaccines-10-00083-f001:**
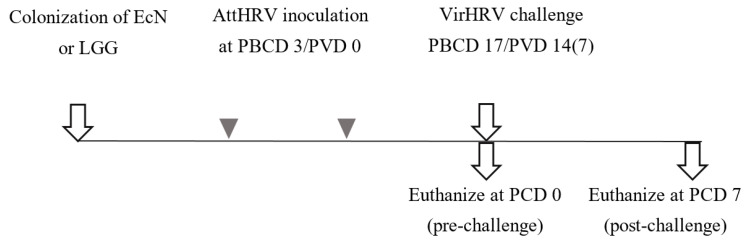
Experimental design showing timepoints for colonization of *Escherichia coli* Nissle (EcN) 1917, *Lacticaseibacillus rhamnosus* GG (LGG) probiotics, live oral attenuated human rotavirus (AttHRV) vaccination, virulent human rotavirus (VirHRV) challenge, and euthanasia. Post bacterial colonization day, PBCD; post vaccination day, PVD; post challenge, PCD.

**Figure 2 vaccines-10-00083-f002:**
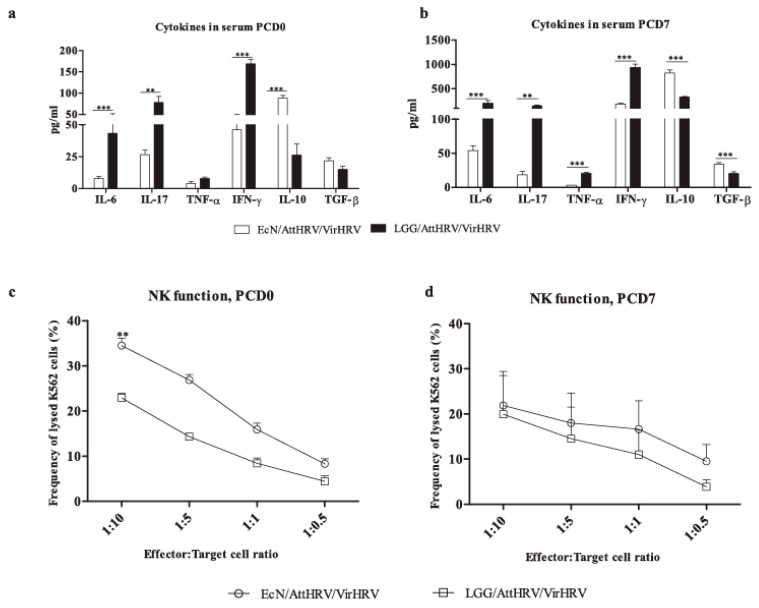
EcN colonization reduced the proinflammatory and increased the immunoregulatory cytokines in serum and modulated the function of natural killer (NK) cells. Mean concentrations of proinflammatory (IL-6, IL-17, TNF-α, IFN-γ) and T regulatory (IL-10, TGF-β) cytokines in serum determined at PCD0 (**a**) and PCD7 (**b**) by ELISA assay. NK cell function in blood MNCs (**c**,**d**). Blood MNCs and carboxyfluorescein diacetate succinimidyl ester (CFSE)-stained K562 tumor cells were used as effector and target cells, respectively, and cocultured at set ratios to assess the NK cytotoxic function. Data are shown as means ± SEM, EcN vs. LGG, and significant differences (** *p* < 0.01, *** *p* < 0.001) are indicated as calculated by nonparametric Kruskal–Wallis rank sum test. Gnotobiotic pigs (Gn) were derived with hysterectomy. EcN and LGG inoculum was inoculated at 3 days of age. All pigs were orally vaccinated twice at a 10 day interval with AttHRV at PBCD 3/post first vaccination day, PVD 0 and PBCD 10/PVD 7(0) (post second vaccination day 0, thereafter referred to as PVD7 (0)). At PBCD 17/PVD 14 (7)/post challenge day (PCD) 0, pigs were euthanized to assess vaccine responses pre challenge. The pigs were challenged with VirHRV and euthanized at PBCD 24/PVD 21 (14)/PCD 7. *Escherichia coli* Nissle 1917, EcN; *Lacticaseibacillus*
*rhamnosus* GG (LGG); post bacterial colonization day, PBCD; live oral attenuated human rotavirus vaccine, AttHRV; virulent human rotavirus, VirHRV.

**Table 1 vaccines-10-00083-t001:** The summary of diarrhea and fecal VirHRV shedding post VirHRV challenge (PCD1–PCD6).

		Diarrhea ^b^				HRV Shedding ^c^			
Group ^a^	*N*	Diarrhea	Mean Cumulative Fecal Score ^d^	Mean Days to Onset of Diarrhea	Mean Duration of Diarrhea ^e^	HRV Shedding	Geometric Mean of Peak Titer Shed (FFU/mL) ^f^	Mean Days to Onset of Virus Shedding	Mean Duration of Viral Shedding
		(%)				(%)			
EcN/AttHRV/VirHRV	7	42.9	6.1	5.3	0.4	42.9	100	3.6	1.1
LGG/AttHRV/VirHRV	6	50	7	4	1	66.7	171	4	0.8

^a^ Gnotobiotic pigs were derived with hysterectomy. Respective pigs were orally vaccinated twice with attenuated human rotavirus (AttHRV) on post bacterial colonization day (PBCD) 3 and 10, challenged with virulent human rotavirus (VirHRV) on PBCD17/PVD14(7)/PCDO, and euthanized on PBCD24/PVD21(14)/PCD7. PVD, post first (second) vaccination day; PCD, post challenge day. ^b^ Pigs with fecal scores > 1 were considered diarrheic. Fecal consistency was scored as follows: 0, normal; 1, pasty; 2, semi-liquid; 3, liquid. ^c^ Determined by cell culture immunofluorescence assay and expressed as FFU/mL. ^d^ Mean of total of fecal score from PCD1–PCD6. ^e^ Mean of the total days with fecal score > 1. ^f^ Samples negative for HRV detection (<25) were assigned a titer of 12.5 for statistical analysis.

**Table 2 vaccines-10-00083-t002:** EcN colonization affected the HRV-specific antibody titers, antibody-secreting cells, and B-cell subpopulations in tissues.

B-Cell Immune Responses		AttHRV × 2 Pre-Challenge		AttHRV × 2 Post VirHRV Challenge	
HRV-specific IgA antibody geometric mean titers		EcN	LGG	EcN	LGG
	Serum	70 (±4)	60 (±30)	1560 * (±1920)	360 (±120)
	SIC	8.5 (±1)	8.5 (±1)	640 * (±256)	28 (±16)
	LIC	u/d	u/d	22 (±16)	8 (±16)
Mean numbers of HRV-specific IgA ASCs	Blood	1.7 (±0)	1.2 (±0)	3.6 (±0)	2.5 (±0)
	Spleen	1.5 (±0)	1 (±0)	1.8 (±0)	0.8 (±0)
	Ileum	77 (±5)	62 (±0)	323 (±10)	157 (±0)
	Duodenum	122 (±50)	83 (±20)	291 (±15)	288 (±0)
Activated antibody-forming B cells (%)	Blood	20 (±16)	25 (±16)	64 (±16)	55 (±7)
	Spleen	10 (±7)	9 (±11)	54 (±13)	43 (±18)
	Ileum	13 (±12)	14 (±19)	39 * (±10)	11 (±12)
	Duodenum	33 (±13)	43 (±12)	62 * (±16)	30 (±7)
Naïve antibody-forming B cells (%)	Blood	2.6 (±1)	3.3 (±2)	3.3 (±4)	2.3 (±1)
	Spleen	4 (±3)	5.6 (±6)	4 * (±3)	2 (±1)
	Ileum	5.6 (±5)	6.6 (±6)	8 *** (±7)	3 (±3)
	Duodenum	5.3 (±4)	4 (±5)	6.2 *** (±6)	2.1 (±3)
Resting/memory antibody-forming B cells (%)	Blood	29 (±25)	35 (±3)	62 (±7)	58 (± 6)
	Spleen	38 (±3)	43 (±4)	36 (±3)	32 (±3)
	Ileum	44 (±5)	50 (±4)	29 (±2)	26 (±2)
	Duodenum	30 (±3)	47 (±4)	33 (±3)	26 (±3)
Ig-secreting B cells (%)	Blood	3.6 (±2)	5.6 (±4)	19 * (±6)	12 (±2)
	Spleen	5 (±4)	9.3 (±2)	53 (±5)	49 (±4)
	Ileum	13 (±1)	27 * (±5)	55 (±5)	50 (±5)
	Duodenum	5.3 (±3)	17 *** (±8)	35 * (±3)	18 (±7)
IgA^+^ B cells (%)	Blood	1.8 (±2)	2 (±1)	7.7 * (±5)	3.7 (±1)
	Spleen	2.3 (±1)	3.1 (±1)	6 (±3)	3.7 (±2)
	Ileum	3 (±2)	3.5 (±8)	10 * (±2)	5 (±1)
	Duodenum	3.6 (±1)	3.5 (±2)	16 (±2)	12 (±3)

Gnotobiotic pigs (Gn) were derived with hysterectomy. EcN and LGG colonization was done at 3 days of age. All pigs were orally vaccinated twice at a 10 day interval with AttHRV at PBCD 3/post first vaccination day, PVD 0 and PBCD 10/PVD 7(0) (post second vaccination day 0, thereafter referred to as PVD7 (0)). At PBCD 17/PVD 14 (7)/post challenge day (PCD) 0, pigs were euthanized to assess vaccine responses pre challenge. The pigs were challenged with VirHRV and euthanized at PBCD 24/PVD 21 (14)/PCD 7. HRV-specific IgA antibodies in serum, small intestinal contents (SIC), and large intestinal contents (LIC) were determined by ELISA and expressed in geometric mean titers. Mean numbers of HRV specific IgA antibody secreting cells (ASCs) per 10^5^ MNCs were determined in systemic and intestinal tissues. Frequencies of activated antibody-forming B cells (CD79β^+^CD2^+^CD21^−^), naïve antibody-forming B cells (CD79β^+^CD2^+^CD21^+^), resting/memory antibody-forming B cells (CD79β^+^CD2^−^CD21^−^), Ig-secreting B cells (CD79β^+^CD2^−^CD21^+^), and IgA^+^ B cells (CD79β^+^IgA^+^) were determined in systemic and intestinal tissues by flow cytometry (%). Data are shown as means ± SEM, EcN vs. LGG, and significant differences (* *p* < 0.05, *** *p* < 0.001) are indicated as calculated by nonparametric Kruskal–Wallis rank sum test. *Escherichia coli* Nissle 1917, EcN; *Latcicaseibacillus rhamnosus* GG (LGG); post bacterial colonization day, PBCD; live oral attenuated human rotavirus vaccine, AttHRV; virulent human rotavirus, VirHRV; undetectable, u/d.

**Table 3 vaccines-10-00083-t003:** EcN colonization altered the frequencies of T helper cells, cytotoxic T cells, HRV-specific IFN-γ producing T cells, CD4^+^/CD8^+^ activated, inducible, and natural T regulatory cells in tissues.

T Cells Immune Responses		AttHRV × 2 Pre-Challenge		AttHRV × 2 Post-VirHRV Challenge	
T helper cells (%)		EcN	LGG	EcN	LGG
	Blood	11 (±5)	15 (±6)	7.7 (±1)	15 (±4)
	Spleen	11 (±5)	12 (±3)	9.1 (±4)	10 (±2)
	Ileum	5.3 (±7)	3.7 (±2)	4.5 (±3)	6.3 (±2)
	Duodenum	4.4 (±7)	3.8 (±5)	12.5 (±4)	10.6 (±5)
Cytotoxic T cells (%)	Blood	10 (±4)	8.3 (±2)	12.5 (±5)	14 (±2)
	Spleen	8.5 (±3)	7.3 (±2)	10.2 (±9)	8.3 (±3)
	Ileum	11.7 (±4)	8.0 (±6)	11.5 (±8)	15 (±6)
	Duodenum	20 (±5)	21 (±7)	38 (±2)	30 (±6)
CD4^+^ IFNγ T cells (%)	Spleen	2.9 (±7)	3.1 (±1)	8.4 (±4)	8.4 (±3)
	Ileum	1.9 (±0)	2.3 (±0)	11.3 (±6)	5.5 (±5)
CD8^+^ IFNγ T cells (%)	Spleen	8.4 (±9)	6 (±3)	7.5 (±5)	11.5 (±7)
	Ileum	2.2 (±0)	6.6 (±8)	4.6 (±2)	2.8 (±2)
Activated CD4^+^ T regs (%)	Blood	4.3 (±5)	4.2 (±3)	3.4 (±3)	3.7 (±7)
	Spleen	7.2 (±2)	6.4 (±4)	9.9 (±9)	6.8 (±12)
	Ileum	17 (±4)	16 (±5)	12 (±14)	9.2 (±12)
	Duodenum	22 (±5)	15 (±6)	31 * (±11)	9.8 (±16)
Activated CD8^+^ T regs (%)	Blood	7.0 (±9)	5.8 (±9)	2.9 (±3)	2.9 (±2)
	Spleen	9.0 (±11)	6.7 (±13)	7.7 (±8)	6.1 (±9)
	Ileum	19 (±22)	18 (±19)	9.6 (±12)	9.7 (±13)
	Duodenum	18 (±12)	32 (±35)	16 (±21)	14 (±21)
Inducible CD4^+^ T regs (%)	Blood	4.7 (±12)	0.7 (±0)	8.8 (±1)	5.7 (±0)
	Spleen	6.1 (±8)	4.3 (±8)	14 (±1)	6.9 (±3)
	Ileum	17 (±23)	2.5 (±2)	6.7 (±1)	16 (±2)
	Duodenum	14 (±31)	7.6 (±11)	8.6 (±0)	8.2 (±0)
Inducible CD8^+^ T regs (%)	Blood	1.8 (±2)	0.4 (±0)	1.3 (±0)	1.8 (±0)
	Spleen	0.8 (±2)	0.8 (±1)	2.8 (±0)	1.8 (±0)
	Ileum	3.0 (±3)	1.9 (±2)	0.9 (±0)	3.6 (±0)
	Duodenum	3.7 (±7)	2.9 (±3)	2.5 (±0)	2.5 (±0)
Natural CD4^+^ T regs (%)	Blood	3.4 (±5)	3.4 (±5)	1.1 (±0)	1.4 (±0)
	Spleen	4.3 (±5)	3.3 (±6)	8.2 (±3)	4.2 (±6)
	Ileum	6.5 (±6)	3.2 (±0)	4.8 (±2)	3.4 (±3)
	Duodenum	10 (±17)	11 (±11)	6.2 (±4)	5.6 (±2)
Natural CD8^+^ T regs (%)	Blood	0.5 (±0)	0.6 (±0)	0.2 (±0)	0.1 (±0)
	Spleen	0.8 (±1)	0.5 (±0)	0.8 (±0)	0.3 (±0)
	Ileum	3.1 ** (±4)	1.3 (±1)	0.9 (±0)	0.7 (±0)
	Duodenum	2.0 (±3)	2.3 (±4)	1.2 (±1)	0.8 (±0)

Gnotobiotic pigs (Gn) were derived with hysterectomy. EcN and LGG colonization was done at 3 days of age. All pigs were orally vaccinated twice at a 10 day interval with AttHRV at PBCD 3/post first vaccination day, PVD 0 and PBCD 10/PVD 7(0) (post second vaccination day 0, thereafter referred to as PVD7 (0)). At PBCD 17/PVD 14 (7)/post challenge day (PCD) 0, pigs were euthanized to assess vaccine responses pre challenge. The pigs were challenged with VirHRV and euthanized at PBCD 24/PVD 21 (14)/PCD 7. Mean frequencies of T helper cells (CD3^+^CD4^+^), cytotoxic T cells (CD3^+^CD8^+^), HRV-specific CD3^+^CD4/CD8^+^ IFNγ-producing T cells, activated (CD4^+^/CD8^+^CD25^+^FOXP3−) T regs, inducible (CD4^+^/CD8^+^CD25^−^FOXP3^+^) T regs, and natural (CD4^+^/CD8^+^CD25^+^FOXP3^+^) T regs were determined in systemic and intestinal tissues by flow cytometry (%). Data are shown as the mean ± SEM, EcN vs. LGG, and significant differences (* *p* < 0.05, ** *p* < 0.01) are indicated as calculated by nonparametric Kruskal–Wallis rank sum test. *Escherichia coli* Nissle 1917, EcN; *Latcicaseibacillus rhamnosus* GG (LGG); post bacterial colonization day, PBCD; live oral attenuated human rotavirus vaccine, AttHRV; virulent human rotavirus, VirHRV; T regulatory cells, T regs.

**Table 4 vaccines-10-00083-t004:** EcN colonization altered the frequencies of cDCs, pDCs, CD103^+^ cDCs/pDCs, activated cDCs/pDCs, Toll like receptor-expressing mononuclear cells, natural killer cell frequency, and apoptotic MNCs in tissues.

Innate Immune Responses		AttHRV × 2 Pre Challenge		AttHRV × 2 Post VirHRV Challenge	
cDCs (%)		EcN	LGG	EcN	LGG
	Blood	3.2 (±4)	3.8 (±3)	3.4 (±2)	2.6 (±2)
	Spleen	3.7 (±2)	2.7 (±3)	5.4 (±2)	4.7 (±3)
	Ileum	2 * (±0)	1.2 (±1)	1.7 * (±1)	0.7 (±0)
	Duodenum	4.8 (±5)	1.8 (±2)	2.6 (±1)	2.1 (±3)
pDCs (%)	Blood	2.4 (±1)	1.7 (±2)	5.1 (±5)	3.1 (±1)
	Spleen	2.8 (±2)	2.6 (±2)	2.0 (±1)	1.8 (±0)
	Ileum	5.6 (±4)	5.1 (±5)	4.8 (±2)	4.2 (±3)
	Duodenum	3.4 (±4)	5.7 (±4)	3.3 (±3)	4.1 (±2)
CD103^+^ cDCs (%)	Blood	1.3 (±1)	1.5 (±1)	1.1 (±0)	1.9 (±0)
	Spleen	0.6 (±0)	1.0 (±1)	1.6 (±0)	0.8 (±1)
	Ileum	2.9 (±7)	1.4 (±1)	6.4 (±19)	1.0 (±0)
	Duodenum	2.4 (±1)	1.0 (±1)	1.7 (±2)	1.6 (±2)
CD103^+^ pDCs (%)	Blood	1.3 (±1)	1.5 (±1)	1.6 (±0)	1.7 (±4)
	Spleen	0.6 (±0)	1.0 (±1)	1.2 (±0)	1.0 (±1)
	Ileum	3.0 (±7)	1.4 (±1)	1.4 (±0)	1.8 (±2)
	Duodenum	2.4 (±1)	1.0 (±1)	1.5 (±0)	1.4 (±3)
Activated cDCs (%)	Blood	1.3 (±1)	1.5 (±1)	3.6 (±7)	3.1 (±4)
	Spleen	0.6 (±0)	1.0 (±1)	2.4 (±1)	2.1 (±2)
	Ileum	3.0 (±7)	1.4 (±1)	2.9 (±7)	4.5 (±7)
	Duodenum	2.4 (±1)	1.0 (±1)	5.9 (±8)	6.2 (±8)
Activated pDCs (%)	Blood	0.8 (±0)	1.4 (±2)	2.3 (±1)	1.3 (±0)
	Spleen	0.5 (±0)	0.9 (±0)	1.4 (±0)	1.7 (±2)
	Ileum	0.7 (±1)	1.6 (±2)	4.9 (±5)	6.4 (±4)
	Duodenum	2.7 (±5)	1.6 (±2)	4.7 (±9)	7.4 (±9)
TLR4 (%)	Blood	2.0 (±2)	2.5 (±2)	1.1 (±0)	3.4 (±9)
	Spleen	2.9 (±3)	3.5 (±4)	1.5 (±1)	1.9 (±2)
	Ileum	3.4 (±3)	5.9 (±3)	1.6 (±0)	2.3 (±0)
	Duodenum	3.9 (±4)	6.9 (±5)	3.0 (±0)	3.5 (±0)
TLR3 (%)	Blood	5.8 (±11)	0.4 (±0)	2.5 (±0)	0.6 (±0)
	Spleen	4.9 ** (±5)	1.1 (±1)	2.1 (±4)	2.9 (±0)
	Ileum	4.0 (±0)	2.9 (±4)	6.5 (±12)	4.7 (±6)
	Duodenum	4.1 (±3)	2.2 (±2)	4.2 (±7)	3.5 (±3)
TLR9 (%)	Blood	3.6 (±4)	0.4 (±0)	0.6 (±0)	1.2 (±0)
	Spleen	5.1 ** (±5)	1.1 (±1)	0.4 (±0)	0.6 (±0)
	Ileum	4.0 (±0)	2.9 (±4)	3.7 (±5)	3.5 (±5)
	Duodenum	3.3 (±3)	2.2 (±2)	2.9 (±4)	4.4 (±16)
NK Frequency (%)	Blood	39 (±43)	30 (±38)	51 (±28)	38 (±40)
	Spleen	7.3 (±6)	12 ** (±13)	13 (±11)	12 (±17)
	Ileum	13 (±5)	4.4 (±3)	2.5 (±1)	4.2 (±2)
	Duodenum	25 (±28)	28 (±16)	12 (±5)	11 (±10)
Apoptotic MNCs (%)	Blood	5.8 (±8)	5.5 (±5)	2.9 (±3)	4.3 (±5)
	Spleen	11 (±14)	5.3 (±5)	4.2 (±4)	4.5 (±8)
	Ileum	2.9 (±3)	4.5 (±7)	3.4 (±6)	4.5 (±8)
	Duodenum	1.3 (±1)	2.3 (±2)	2.6 (±2)	2.4 (±1)

Gnotobiotic pigs (Gn) were derived with hysterectomy. EcN and LGG inoculum colonization was done at 3 days of age. All pigs were orally vaccinated twice at a 10 day interval with AttHRV at PBCD 3/post first vaccination day, PVD 0 and PBCD 10/PVD 7(0) (post second vaccination day 0, thereafter referred to as PVD7 (0)). At PBCD 17/PVD 14 (7)/post challenge day (PCD) 0, pigs were euthanized to assess vaccine responses pre challenge. The pigs were challenged with VirHRV and euthanized at PBCD 24/PVD 21 (14)/PCD 7. Mean frequencies of cDCs (SWC3a^+^CD4^−^CD11R1^+^), pDCs (SWC3a^+^CD4^+^CD11R1^−^), CD103^+^ cDCs (SWC3a^+^CD4^−^), CD103^+^ pDCs (SWC3a^+^CD4^+^), activated cDCs (SWC3a^+^CD4^−^CD11R1^+^MHC II^+^), activated pDCs (SWC3a^+^CD4^+^CD11R1^−^MHC II^+^), Toll like receptor (TLR)-expressing mononuclear cells (MNCs), natural killer (NK) cells, and apoptotic MNCs were determined by flow cytometry (%). Data are shown as means ± SEM, EcN vs. LGG, and significant differences (* *p* < 0.05, ** *p* < 0.01) are indicated as calculated by nonparametric Kruskal–Wallis rank sum test. *Escherichia coli* Nissle 1917, EcN; *Latcicaseibacillus rhamnosus* GG (LGG); post bacterial colonization day, PBCD; live oral attenuated human rotavirus vaccine, AttHRV; virulent human rotavirus, VirHRV.

## Data Availability

The data presented in this study are available on request from the corresponding author.

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
