# Peer review of "Escherichia coli Nissle 1917 Enhances Efficacy of Oral Attenuated Human Rotavirus Vaccine in a Gnotobiotic Piglet Model"

_vaccines, 2022, doi:10.3390/vaccines10010083_

Round 1

Reviewer 1 Report

  1. I do not see Table 1 in the manuscript I downloaded. Please add it to the paper.
  2. Please provide a brief description of the methods used for colonizing Gn piglets. This will help clarify the procedure for readers.
  3. In line 201, the authors state although EcN colonization decreased viral HRV shedding, the mean number of days with HRV shedding was higher than the LGG group. Does this suggest that EcN may result in sustained low-level transmission? While decreased viral shedding implies impaired transmission, an elongated timeframe of viral shedding may allow for increased number of exposures to bystanders.
  4. The authors repeatedly attribute priming of the adaptive immune response to AttHRV. Is it reasonable to assume that priming is at least partially attributable to colonization? I would think EcN or LGG colonization would serve as a primer for, at a minimum, T cell immunity?
  5. In line 312 the authors state “resulted in significantly in most cases reduced”. Please clarify this statement.
  6. In section 3.10, the authors show decreased NK cell activity in EcN colonized pigs following HRV challenge (Figure 2D). Is the belief that this occurs due to rapid clearance of the virus? If so, why does NK cell activity decrease following challenge compared to the vaccination phase? What happens to the activated NK cells? Have they all been recruited to the intestine? Why does this decrease activity of the circulating NK cell population?
  7. How do the authors envision the vaccine regimen? The discussion suggests that EcN will be given with AttHRV. Would it be more effective to give EcN a few weeks prior to AttHRV?

Author Response

Review Report (Reviewer # 1)

RE -1: I do not see Table 1 in the manuscript I downloaded. Please add it to the paper.

AU: When manuscript was uploaded to the Vaccines website, table 1 was visible in both MS Word and PDF copies. However, when I received copy from the editor with MDPI logo peer review copy, table 1 wasn’t available. Table 1 may have omitted accidentally during the conversion process. We apologize for inconvenience. Hopefully table 1 will show when the  revised version of the manuscript is uploaded. Table 1 is in lines 225-238.  

RE -2: Please provide a brief description of the methods used for colonizing Gn piglets. This will help clarify the procedure for readers.

AU: A brief description of the methods used for colonizing Gn piglets is now added, lines 129-130.

RE -3: In line 201, the authors state although EcN colonization decreased viral HRV shedding, the mean number of days with HRV shedding was higher than the LGG group. Does this suggest that EcN may result in sustained low-level transmission? While decreased viral shedding implies impaired transmission, an elongated timeframe of viral shedding may allow for increased number of exposures to bystanders.

AU: We have revised this statement to say that there was no difference as the HRV shedding was only marginally and not significantly higher in the EcN group (1.1 vs. 0.8 days). Not even half a day longer, lines 244-246. Considering lower shedding titters and similar shedding duration, there is no evidence to support the hypothesis that EcN may support sustained low-level transmission.

RE -4: The authors repeatedly attribute priming of the adaptive immune response to AttHRV. Is it reasonable to assume that priming is at least partially attributable to colonization? I would think EcN or LGG colonization would serve as a primer for, at a minimum, T cell immunity?

AU: Yes, the reviewer’s suggestion is correct. EcN and LGG colonization prime the innate (line 575) and T cell immunity (line 543) .

RE -5: In line 312 the authors state “resulted in significantly in most cases reduced”. Please clarify this statement.

AU: The sentence is modified now and highlighted, line 389.

RE -6: In section 3.10, the authors show decreased NK cell activity in EcN colonized pigs following HRV challenge (Figure 2D). Is the belief that this occurs due to rapid clearance of the virus? If so, why does NK cell activity decrease following challenge compared to the vaccination phase? What happens to the activated NK cells? Have they all been recruited to the intestine? Why does this decrease activity of the circulating NK cell population?

AU: In fact, we show that the NK activity was increased in the EcN colonized pigs, section 3.10/Figures 2C and 2D

RE -7: How do the authors envision the vaccine regimen? The discussion suggests that EcN will be given with AttHRV. Would it be more effective to give EcN a few weeks prior to AttHRV?

AU: Our results indicate that EcN colonization has been efficacious when done three days prior to vaccination. We have revised the text to acknowledge that (lines 595-597).

Reviewer 2 Report

This study is very useful for enhance protection against human rotavirus infection by live attenuated oral vaccine. However, it will better to be revised bellow for understand benefit of using live attenuated oral human rotavirus vaccine.

  1. In introduction; Please mention the effectiveness of probiotics enhance efficacy not only human vaccines against viral gastroenteritis, but also piglet and calf vaccine against viral diarrhea.
  2. In discussion; Please explain why the results of this study and previous study have been appeared.
  3. In discussion; Please explain more clearly, the mechanisms of differences between Gram positive LGG probiotic, and Gram negative EcN probiotics.

Author Response

Review Report (Reviewer # 2)

RE -1: In introduction; Please mention the effectiveness of probiotics enhance efficacy not only human vaccines against viral gastroenteritis, but also piglet and calf vaccine against viral diarrhea.

AU: Above sentence is inserted into the introduction and highlighted, line 105.

RE -2: In discussion; Please explain why the results of this study and previous study have been appeared.

AU: Now it has been explained and highlighted, lines 509-514.

RE -3: In discussion; Please explain more clearly, the mechanisms of differences between Gram positive LGG probiotic, and Gram negative EcN probiotics.

AU: Mechanisms of EcN and LGG are now explained and highlighted, lines 578-585.